# Prevalence of dyslipidemia and associated risk factors among adult residents of Mekelle City, Northern Ethiopia

**Gebremedhin Gebreegziabiher** [1]*, **Tefera Belachew**[2], **Kibrti Mehari**[3], **Dessalegn Tamiru**[2]

**1** Department of Human Nutrition, School of Public Health, College of Medicine and Health Sciences, Adigrat University, Adigrat, Tigray, Ethiopia, **2** Department of Nutrition and Dietetics, Faculty of Public Health, Jimma University, Jimma, Oromia, Ethiopia, **3** Tigray Health Research Institute, Mekelle, Tigray, Ethiopia

* ghingherg@gmail.com

## Abstract

### Introduction

Dyslipidemia is a major risk factor for cardiovascular diseases (CVD). The prevalence of dyslipidemia is not known among Ethiopian adults. The prevalence is expected to rise due to the socio-economic development accompanied by lifestyle changes. This study was conducted to estimate the prevalence of dyslipidemia and associated risk factors among adult residents of Mekelle City.

### Methods

A community-based cross-sectional study was conducted among 321 randomly selected subjects. Data were collected on sociodemographic, anthropometric, lifestyle, and clinical characteristics of the participants using the WHO STEPS survey instrument. Data were analyzed using SPSS software version 24.0. Student's t-test and Pearson's Chi-square test were used to assessing the interrelationship between each factor and outcome variables. Bivariate and multivariable logistic regression analysis were used to identify risk factors associated with dyslipidemia. All statistical significance was considered at p ≤0.05.

### Results

The prevalence of dyslipidemia in this study was 66.7%. The prevalence of high low-density lipoprotein cholesterol (LDL-C), elevated triglyceride, elevated total cholesterol, and low high-density lipoprotein cholesterol (HDL-C) was 49.5%, 40.2%, 30.8%, and 16.5%, respectively. Being above 64 years (aOR: 2.196, 95% CI: 1.183–4.078) and 40–64 years old (aOR: 2.196, 95% CI: 1.183–4.078), overweight (aOR: 2.50, 95% CI: 1.314–4.756) and obesity (aOR: 15.489, 95% CI: 3.525–68.070), walking <150 minutes per week (aOR: 1.722, 95% CI: 1.004–2.953), raised fasting blood glucose (FBG) (aOR: 4.804, 95% CI: 1.925–11.988), and medium socio-economic status (aOR: 2.017, 95% CI: 1.044–3.899) were identified as significant predictors of dyslipidemia.

**Data Availability Statement:** All relevant data are within the manuscript and its Supporting Informationw files.

**Funding:** The award was received by GGG. No grant number for the award. website of Jimma University: www.ju.edu.et Website of Adigrat University: http://adu.edu.et.usitestat.com/ Jimma University and Adigrat Universities have funded this research in collaboration. The funders had no role in study design, data collection and analysis, decision to publish, or preparation of the manuscript.

**Competing interests:** The authors have declared that no competing interests exist.

## Conclusions

The finding of this study indicated that the prevalence of dyslipidemia is unacceptably high among adult residents of Mekelle City, which underlines an urgent need for early detection and public health interventions through the integrated involvement of public, governmental, and non-governmental organizations.

## Introduction

Dyslipidemia is either one or a combination of elevated total cholesterol, high LDL-C, low HDL-C, and elevated triglyceride [1]. Dyslipidemia is a major risk factor for coronary heart disease (CHD) [2]. People with dyslipidemia are at a twofold increased risk of developing CVD as compared to those with normal lipid levels [3]. CVD is becoming more prevalent globally and is one of the prominent causes of death [4]. Raised levels of certain lipids in the blood increase the risk of atherosclerosis, which is recognized as the primary risk factor for stroke, peripheral vascular, and CHD [5].

Both LDL-C and HDL-C regulate the amount of cholesterol in the body. An imbalance between the two can increase the risk of myocardial infarction and stroke. High LDL-C is associated with an increased risk of atherosclerotic CVD due to the buildup of plaques within the arteries [1]. LDL-C carried cholesterol is potentially atherosclerotic. However, the HDL-C carried one has a protecting role against atherosclerosis [6]. HDL-C helps to remove cholesterol from the body, which decreases the risk of atherosclerotic CVD [1].

Most (80%) of the lipid disorders are associated with diet and lifestyle [7]. Modifiable risk factors, including a diet high in saturated or trans fats, sedentary lifestyle, smoking, and obesity increase the risk of dyslipidemia [3]. Lipid profile and CVD have a linear relationship. Dyslipidemia aggravates the development of atherosclerosis [8]. The prevalence of dyslipidemia is much higher among patients with coexisting cardiovascular risk factors such as hypertension, diabetes, or human immunodeficiency virus [9].

Dyslipidemia is associated with more than half of the global cases of ischemic heart disease and more than 4 million deaths per annum [10]. The pooled prevalence of dyslipidemia among the African general adult population was 25.5%. Besides, the overall prevalence of elevated total cholesterol, high LDL-C, low HDL-C, and elevated triglyceride was 25.5%, 21.4%, 19.5%, and 17.0%, respectively [9]. Advanced age, raised FBG, drinking coffee, and vegetable intake were identified as significant predictors of dyslipidemia among women contraceptive users in Eastern Ethiopia [11].

There is no literature showing the prevalence and factors associated with dyslipidemia among the Ethiopian general adult population. Only one facility-based cross-sectional study was conducted in Eastern Ethiopia among contraceptive users, which reported a high prevalence (34.8%) of dyslipidemia. However, it cannot represent the general adult population [11]. Therefore, this study aimed to assess the prevalence of dyslipidemia and associated risk factors among adult residents in Mekelle City, Northern Ethiopia.

## Materials and methods

### Study design, setting, and population

A community-based cross-sectional study was conducted among 321 adult residents in Mekelle City from July to September 2019. Mekelle is the second-largest city in Ethiopia, which is the

capital city of Tigray regional state. Mekelle is located at 783 km to the north of the capital city, Addis Ababa. The city is divided into seven sub-cities. Being an adult aged 20 years and above and residents who lived at least 6 months in the city were considered as the inclusion criteria. Whereas, pregnant women and the first 6 months of lactating mothers were excluded from the study.

## Ethical issues

Ethical clearance and approval were obtained from the institutional review board of Jimma University. Besides, a support letter was obtained from the local administrators. All participants were informed of what is expected from them and their rights. Written informed consent was obtained from each participant. Illiterate participants put their fingerprint as a signature in the written consent form voluntarily after data collectors read the information. Participants with abnormal lipid profiles, FBG, and blood pressure were linked to their nearby healthcare facilities for further investigation, counseling, and treatment.

## Sample size determination and sampling procedure

A single population proportion formula was used [12] to determine the sample size with the assumption of the prevalence of dyslipidemia among the African adults (25.5%) [9], 95% level of confidence, 5% margin of error, and 80% power. Thus, the calculated sample size was 292. After 10% (n = 29) of the calculated sample size was added to consider the non-response rate, the total sample size was 321.

Probability of having dyslipidemia increases with age. Hence, a stratified sampling technique was used using age. Two age groups (strata) were created (20–39 years and ≥40 years). Then, the total sample size was proportionally allocated based on the number of the adult population in each stratum. Besides, the total sample size was proportionally allocated to the seven sub-cities. The list of households from each municipality was used as a sampling frame. Households were selected using a simple random sampling technique from each sub-city. Likewise, a single eligible participant was selected using a lottery method from each selected household.

## Data collection and quality control

A data collection team was established. The team consisted of three public health professionals, one medical laboratory technologist, and one supervisor. The team was trained for two days on how to conduct face-to-face interviews, anthropometric measurements, how to measure blood pressure, and how to collect and handle the blood sample. A structured questionnaire adapted from the WHO STEPS survey instrument was used to collect the data [13]. The questionnaire was translated into the local language (Tigrigna). Besides, it was translated back to English to check the consistency. The data collection tools were pretested to check completeness, consistency, sensitivity, and applicability and were ratified accordingly.

Height was measured using a stadiometer (UNICEF SECA) to the nearest 0.1 cm without shoes. The participant was positioned in the Frankfurt plane and the measurer checked the four contact points (heel, calf, buttock, and shoulder) against the vertical stand. Waist Circumference (WC) was taken to the nearest 0.1 cm at the midway between the lowest costal margin, midclavicular line, and the anterior superior iliac spine using fixed tension tape. Hip circumference was also taken to the nearest 0.1 cm at the level of the greater trochanter of the femur with the subjects wearing a pant. Weight was measured using a digital scale (UNICEF SECA) to the nearest 0.1 kg with light closes and without shoes.

Blood pressure was measured in triplicate after 5 minutes of rest and the subsequent measurements were done 5 minutes apart. The mean systolic blood pressure (SBP) and diastolic blood pressure (DBP) were used for analyses. The validity of the weighing scale was checked

using a known weight before each measurement. All anthropometric measurements were collected in triplicate and the average values were used for analyses. All measurement data were collected using standardized techniques and calibrated equipment [14].

### Sample collection, laboratory analysis, and definition of terms

After conducting the face-to-face interviews and anthropometric measurements, participants were appointed to the next morning (8:00 am–9:00 am) at their nearest health facility to give fasting venous blood. Around 5 ml of venous blood sample was collected after overnight fasting for FBG, HDL-C, triglyceride, LDL-C, and total cholesterol tests. The blood sample was clotted for 30 minutes. Then, the sample was centrifuged for 5 minutes at 4000 revolutions per minute. Around 2.5 ml of pure serum sample was separated to the Nunc tube. The sample was analyzed using Bio-system A25 automated clinical chemistry machine (Spain). Before sample analysis, the machine was checked using controls and blank on a daily basis.

Raised FBG was defined as ≥5.6 mmol/L (≥100 mg/dL) or on diabetes treatment and raised WC was defined as ≥94 cm for men and ≥80 cm for women [15]. Dyslipidemia was defined as a lipid profile that consists of the following abnormalities, either singly or in combination. Elevated total cholesterol ≥5.17 mmol/L (≥200 mg/dL); high LDL-C ≥3.36 mmol/L (≥130 mg/dL); low HDL-C <1.03 mmol/L (<40 mg/dL) for men, <1.3 mmol/L (<50 mg/dL) for women; and elevated triglyceride ≥1.7 mmol/L (≥150 mg/dL) [16].

Participants were considered as normotensive (<130/85 mmHg), pre-hypertensive (≥130-139/85-89 mmHg), and hypertensive (≥140/90 mmHg) for SBP and DBP, respectively. BMI was calculated by dividing weight in kilograms (kg) by the square meters (m$^2$) of height. Participants with a BMI lower than 18.5 kg/m$^2$ were considered as underweight; between 18.5 and 24.9 kg/m$^2$ as normal; between 25.0 and 29.9 kg/m$^2$ as overweight and 30.0 kg/m$^2$ and above as obese [17].

### Data analysis

Data were checked for completeness and consistency in the hard copy, double entered into EPI data software version 3.1 to check clerical errors. Then, the data were exported to the Statistical Package for the Social Sciences (SPSS) for Windows version 24 program for analyses. A descriptive analysis of the background characteristics was performed. Besides, the normality of the continuous variables was checked. Bivariate and multivariable logistic regression analyses were performed to identify factors independently associated with dyslipidemia. Backward stepwise elimination was used to remove non-significant variables until only statistically significant variables remained in the final logistic model. Crude and adjusted odds ratios and their corresponding 95% Confidence Intervals (CI) were computed in the bivariate and multivariable logistic regression analysis, respectively. The goodness of fit of the model was checked using the Hosmer-Lemeshow test at p >0.05. All statistical significance was declared at p ≤0.05.

## Results

### Characteristics of the participants

A total of 321 adults participated in the study. Men and women were significantly different in terms of educational status, marriage, occupation, smoking, and alcohol consumption (p <0.005). A higher percentage of women were ever been measured their blood pressure and blood glucose (p <0.005). Whereas, a higher percentage of men did formal exercise and walked ≥150 minutes per week (p <0.005). Only 13.1% of the participants did formal exercise. The intensity of activity in daily work was significantly different across gender (p = 0.044). Women consumed significantly more vegetables per week compared to men (P = 0.007) (Table 1).

**Table 1. Background characteristics of the participants stratified by gender (n = 321).**

| Variables | Categories | Men, n (%) | Women, n (%) | Total, n (%) | p-value |
|---|---|---|---|---|---|
| | | **145 (45.2)** | **176 (54.8)** | **321 (100.0)** | **(X²)** |
| **Age (mean ± SD)** | | 39.98±14.52 | 38.18±13.96 | 38.99±14.22 | 0.259[¶] |
| **Educational status** | Unable to read and write | 6 (4.1) | 29 (16.5) | 35 (10.9) | 0.002* |
| | Primary school | 20 (13.8) | 31 (17.6) | 51 (15.9) | |
| | Secondary school | 70 (48.3) | 64 (36.4) | 134 (41.7) | |
| | Tertiary | 49 (33.8) | 52 (29.5) | 101 (31.5) | |
| **Marital status** | Single | 49 (33.8) | 30 (17.0) | 79 (24.6) | 0.001* |
| | Married | 91 (62.8) | 133 (75.6) | 224 (69.8) | |
| | Others [a] | 5 (3.4) | 13 (7.4) | 18 (5.6) | |
| **Occupation** | Employed | 125 (86.2) | 75 (42.6) | 200 (62.3) | <0.001* |
| | Housewife | 0 (0.0) | 79 (44.9) | 79 (24.6) | |
| | Unemployed | 20 (13.8) | 22 (12.5) | 42 (13.1) | |
| **Household monthly income (ranked)** | Poor | 61 (42.1) | 61 (34.7) | 122 (38.0) | 0.067 |
| | Medium | 42 (29.0) | 42 (23.9) | 84 (26.2) | |
| | Rich | 42 (29.0) | 73 (41.5) | 115 (35.8) | |
| **Smoking** | Yes | 16 (11.0) | 0 (0.0) | 16 (5.0) | <0.001* |
| | No | 129 (89.0) | 176 (100.0) | 305 (95.0) | |
| **Living with smoker** | Yes | 9 (6.2) | 7 (4.0) | 16 (5.0) | 0.361 |
| | No | 136 (93.8) | 169 (96.0) | 305 (95.0) | |
| **Alcohol consumption** | Yes | 123 (84.8) | 125 (71.0) | 248 (77.3) | 0.003* |
| | No | 22 (15.2) | 51 (29.0) | 73 (22.7) | |
| **Ever measured blood pressure** | Yes | 75 (51.7) | 118 (67.0) | 193 (60.1) | 0.005* |
| | No | 70 (48.3) | 58 (33.0) | 128 (39.9) | |
| **Ever told having hypertension** | Yes | 12 (8.3) | 20 (11.4) | 32 (10.0) | 0.358 |
| | No | 133 (91.7) | 156 (88.6) | 289 (90.0) | |
| **On treatment for hypertension** | Yes | 5 (3.4) | 5 (2.8) | 10 (3.1) | 0.759[b] |
| | No | 140 (96.6) | 171 (97.2) | 311 (96.9) | |
| **Ever measured blood glucose** | Yes | 57 (39.3) | 100 (56.8) | 157 (48.9) | 0.002* |
| | No | 88 (60.7) | 76 (43.2) | 164 (51.1) | |
| **Ever told having diabetes** | Yes | 8 (5.5) | 7 (4.0) | 15 (4.7) | 0.515 |
| | No | 137 (94.5) | 169 (96.0) | 306 (95.3) | |
| **On treatment for diabetes** | Yes | 1 (0.7) | 3 (1.7) | 4 (1.2) | 0.630[b] |
| | No | 144 (99.3) | 173 (98.3) | 317 (98.8) | |
| **Formal exercise** | Yes | 38 (26.2) | 4 (2.3) | 42 (13.1) | <0.001* |
| | No | 107 (73.8) | 172 (97.7) | 279 (86.9) | |
| **Walking minutes per week** | None | 6 (4.1) | 20 (11.4) | 26 (8.1) | <0.001* |
| | 1–149 | 18 (12.4) | 87 (49.4) | 105 (32.7) | |
| | ≥150 | 121 (83.5) | 69 (39.2) | 190 (59.2) | |
| **Intensity of activity of daily work** | Low | 15 (10.3) | 17 (9.7) | 32 (10.0) | 0.044* |
| | Moderate | 109 (75.2) | 148 (84.1) | 257 (80.0) | |
| | Vigorous | 21 (14.5) | 11 (6.2) | 32 (10.0) | |
| **Fruit intake per week (mean ± SD)** | | 1.55±2.16 | 1.53±1.77 | 1.54±1.95 | 0.915[¶] |
| **vegetable intake per week (mean ± SD)** | | 1.43±1.48 | 1.90±1.60 | 1.69±1.56 | 0.007*[¶] |

**Note:** *significant difference,

[a] Divorced, Widowed, Separated,

[b] Fisher's Exact Test,

[¶] student's t-test, X²: Chi-square, SD: Standard Deviation.

## Prevalence of dyslipidemia and other CVD risk factors

The prevalence of dyslipidemia in this study was 66.7%. A higher percentage of men had elevated triglyceride (P <0.001). Whereas, a higher percentage of women had low HDL-C, central obesity, and raised waist to hip ratio (P <0.01). The mean waist to hip ratio was significantly higher among men (p = 0.001). The mean values of total cholesterol and HDL-C were significantly higher among women (p<0.001). Whereas, the mean triglyceride was significantly higher among men (p<0.001) (Table 2).

**Table 2. Prevalence of dyslipidemia and other CVD risk factors with their mean values stratified by gender (n = 321).**

| Variables | Categories | Prevalence | | | | Mean ± SD | | | |
|---|---|---|---|---|---|---|---|---|---|
| | | All | Men | Women | p-value[g] | All | Men | Women | p-value[h] |
| | | | n (%) | n (%) | | | | | |
| **Dyslipidemia** | Yes | 214 (66.7) | 99 (68.3) | 115 (65.3) | 0.579 | NA | NA | NA | NA |
| | No | 107 (33.3) | 46 (31.7) | 61 (34.7) | | | | | |
| **Total cholesterol** | <200 mg/dL | 222 (69.2) | 103 (71.0) | 119 (67.6) | 0.509 | 182.8±49.1 | 176.6±45.4 | 187.9±51.6 | 0.04* |
| | ≥200 mg/dL | 99 (30.8) | 42 (29.0) | 57 (32.4) | | | | | |
| **Triglyceride** | <150 mg/dL | 192 (59.8) | 71 (49.0) | 121 (68.8) | <0.001* | 165.5±138.8 | 200.4±161.3 | 136.7±109.4 | <0.001* |
| | ≥150 mg/dL | 129 (40.2) | 74 (51.0) | 55 (31.2) | | | | | |
| **LDL-C** | <130 mg/dL | 162 (50.5) | 80 (55.2) | 82 (46.6) | 0.126 | 144.4±48.2 | 138.4±48.7 | 149.1±47.4 | 0.051 |
| | ≥130 mg/dL | 159 (49.5) | 65 (44.8) | 94 (53.4) | | | | | |
| **HDL-C (M/W)** | <40/50 mg/dL | 53 (16.5) | 15 (10.3) | 38 (21.6) | 0.007* | 57.7±12.7 | 53.6±10.7 | 61.1±13.3 | <0.001* |
| | ≥40/50 mg/dL | 268 (83.5) | 130 (89.7) | 138 (78.4) | | | | | |
| **FBG** | Normal [a] | 251 (78.2) | 107 (73.8) | 144 (81.8) | 0.221 | 97.4±38.0 | 101.0±38.9 | 94.4±37.2 | 0.124 |
| | Pre-diabetes [b] | 40 (12.5) | 22 (15.2) | 18 (10.2) | | | | | |
| | Diabetes [c] | 30 (9.3) | 16 (11.0) | 14 (8.0) | | | | | |
| **BMI** | Underweight | 26 (8.1)) | 10 (6.9) | 16 (9.1) | 0.138 | 24.4±4.9 | 24.1±4.2 | 24.7±5.3 | 0.261 |
| | Normal | 166 (51.7) | 77 (53.1) | 89 (50.6) | | | | | |
| | Overweight | 87 (27.1) | 45 (31.0) | 42 (23.8) | | | | | |
| | Obese | 42 (13.1) | 13 (9.0) | 29 (16.5) | | | | | |
| **WC (M/W)** | <94/80 cm | 161 (50.2) | 95 (65.5) | 66 (37.5) | <0.001* | 85.6±13.6 | 86.6±12.6 | 84.7±14.4 | 0.212 |
| | ≥94/80 cm | 160 (49.8) | 50 (34.5) | 110 (62.5) | | | | | |
| **Waist to hip ratio (M/W)** | <0.9/0.8 | 87 (27.1) | 53 (36.6) | 34 (19.3) | 0.001* | 0.91±0.10 | 0.92±0.09 | 0.89±0.1 | 0.001* |
| | ≥0.9/0.8 | 234 (72.9) | 92 (63.4) | 142 (80.7) | | | | | |
| **SBP/DBP (mmHg)** | Normotensive [d] | 163 (50.8) | 75 (51.7) | 88 (50.0) | 0.305 | 128.4±20.2/ 81.5 ±11.3 | 129.0±19.8/ 82.7 ±12.2 | 127.8±20.5/ 80.5 ±10.6 | 0.618/ 0.094 |
| | Prehypertension [e] | 60 (18.7) | 22 (15.2) | 38 (21.6) | | | | | |
| | Hypertension [f] | 98 (30.5) | 48 (33.1) | 50 (28.4) | | | | | |

**Note:** *significant difference,

[a] FBG: <100.0 mg/dL,

[b] FBG: 100.0–125.9 mg/dL,

[c] FBG: ≥126.0 mg/dL,

[d] SBP/DBP: <130/85 mmHg,

[e] SBP/DBP: 130-139/85-89 mmHg,

[f] SBP/DBP: ≥140/90 mmHg,

[g] Chi-square test,

[h] Student's t-test.

**Abbreviations**: BMI: Body Mass Index, CVD: Cardiovascular Disease, DBP: Diastolic Blood Pressure, FBG: Fasting Blood Glucose, HDL-C: High-Density Lipoprotein Cholesterol, LDL-C: Low-Density Lipoprotein Cholesterol, M: Men, SBP: Systolic Blood Pressure, WC: Waist Circumference, W: Women.

The risk of dyslipidemia and its component lipid abnormalities, except low HDL-C, were consistently increased with WC, waist to hip ratio, waist to height ratio, BMI, FBG, and blood pressure (p <0.005). Similarly, dyslipidemia, total cholesterol, and high LDL-C were consistently increased with age (p <0.001). Though elevated triglyceride was significantly different with age, the rise was not consistent (p = 0.003). Likewise, dyslipidemia, elevated total cholesterol, and high LDL-C was significantly different with the wealth index, the intensity of activity in daily work, and weekly walking time, respectively (p <0.05). Besides, the risk of low HDL-C was consistently increased with WC, waist to height ratio, and BMI (p <0.05). A higher percentage of non-alcohol consumers had low HDL-C (p = 0.033) (Table 3).

## Effect of dyslipidemia on the mean value of different CVD risk factors

The mean values of LDL-C, total cholesterol, triglyceride, FBG, BMI, WC, SBP, DBP, age, and weight were higher among dyslipidemia positive subjects compared to negatives. The mean FBG (102.8 mg/dL) was higher among subjects with dyslipidemia compared to dyslipidemia negatives (86.7 mg/dL). However, the mean value of HDL-C was higher among dyslipidemia negative subjects compared to positives (Fig 1).

## Correlation of lipid components and other CVD risk factors

The most frequently occurred combination of lipid abnormalities in both sexes was TC+TG +LDL-C followed by TC+LDL-C. The combination of TC+TG+LDL-C was the most common among men followed by TG alone. Whereas, TC+LDL-C was the most common combination among women followed by TC+TG+LDL-C. More than one-fourth (27.1%) of the subjects with dyslipidemia have two lipid abnormalities, while 17.4% of them have three lipid abnormalities. All CVD risk factors were positively correlated with each other except with HDL-C. A strong correlation was observed between LDL-C and total cholesterol (r = 0.83), WC and BMI (r = 0.82) and total cholesterol and triglyceride (r = 0.49). HDL-C was positively correlated only with total cholesterol (r = 0.24) and LDL-C (r = 0.18) (Table 4 and Figs 2 and 3).

## Factors associated with dyslipidemia

In the multivariable logistic regression model, advanced age, higher BMI, walking less than 150 minutes per week, raised FBG, and medium socio-economic status were significantly associated with dyslipidemia (p <0.05). The odds of dyslipidemia was 2.2 (aOR: 2.196, 95% CI: 1.183–4.078) and 4.3 (aOR: 4.334, 95% CI: 1.183–15.877) times higher among 40–64 years and ≥65 years, respectively compared to subjects aged below 40 years. Similarly, the odds of dyslipidemia was 2.5 (aOR: 2.50, 95% CI: 1.314–4.756) and 15.5 (aOR: 15.489, 95% CI: 3.525–68.070) times higher among overweight and obese subjects, respectively compared to normal and underweight subjects. Adults who walked less than 150 minutes per week had a 1.7 (aOR: 1.722, 95% CI: 1.004–2.953) times higher risk of dyslipidemia compared to their counterparts. The likelihood of dyslipidemia among adults with raised FBG was 4.8 (aOR: 4.804, 95% CI: 1.925–11.988) times higher compared to adults with normal FBG. Besides, participants with medium socioeconomic status had a 2.0 (aOR: 2.017, 95% CI: 1.044–3.899) times higher risk of dyslipidemia compared to participants with low socioeconomic status (Table 5).

## Discussion

In this study, a high prevalence of dyslipidemia (66.7%) was found among adults residing in Mekelle city. This high prevalence of dyslipidemia might be attributed to rapid urbanization, improved socioeconomic status, change in dietary habits, decreased physical activity, and

**Table 3. Factors affecting the prevalence of dyslipidemia and its lipid components (n = 321).**

| Variables | Categories | Dyslipidemia (n (%)) | Elevated TC (≥200 mg/dL) (n (%)) | Elevated TG (≥150 mg/dL) (n (%)) | High LDL-C (≥130 mg/dL) (n (%)) | Low HDL-C (M/W<40/50 mg/dL) (n (%)) |
|---|---|---|---|---|---|---|
| **Age** | 20–39 | 107 (56.3) | 41 (21.6) | 63 (33.2) | 75 (39.5) | 26 (13.7) |
| | 40–59 | 81 (80.2) | 40 (39.6) | 54 (53.5) | 59 (58.4) | 21 (20.8) |
| | ≥60 | 26 (86.7) | 18 (60.0) | 12 (40.0) | 25 (83.3) | 6 (20.0) |
| | **p-value** | **<0.001***  | **<0.001*** | **0.003*** | **<0.001*** | **0.258** |
| **Alcohol consumption** | Yes | 169 (68.1) | 80 (32.3%) | 105 (42.3) | 127 (51.2) | 35 (14.1) |
| | No | 45 (61.6) | 19 (26.0) | 24 (32.9) | 32 (43.8) | 18 (24.7) |
| | **p-value** | **0.346** | **0.382** | **0.147** | **0.306** | **0.033*** |
| **WC in cm (M/W)** | <94/80 | 86 (53.4) | 35 (21.7) | 47 (29.2) | 59 (36.6) | 16 (9.9) |
| | ≥94/80 | 128 (80.0) | 64 (40.0) | 82 (51.3) | 100 (62.5) | 37 (23.1) |
| | **p-value** | **<0.001*** | **<0.001*** | **<0.001*** | **<0.001*** | **0.001*** |
| **Waist to hip ratio (M/W)** | <0.9/0.8 | 40 (46.0) | 16 (18.4) | 23 (26.4) | 25 (28.7) | 10 (11.5) |
| | ≥0.9/0.8 | 174 (74.4) | 83 (35.5) | 106 (45.3) | 134 (57.3) | 43 (18.4) |
| | **p-value** | **<0.001*** | **0.003*** | **0.002*** | **<0.001*** | **0.140** |
| **Waist to height ratio (M/W)** | <0.49/0.50 | 49 (42.6) | 20 (17.4) | 26 (22.6) | 36 (31.3) | 10 (8.7) |
| | ≥0.49/0.50 | 165 (80.1) | 79 (38.3) | 103 (50.0) | 123 (59.7) | 43 (20.9) |
| | **p-value** | **<0.001*** | **<0.001*** | **<0.001*** | **<0.001*** | **0.005*** |
| **Body mass index** | Underweight | 9 (34.6) | 4 (15.4) | 4 (15.4) | 6 (23.1) | 3 (11.5) |
| | Normal | 96 (57.8) | 41 (24.7) | 52 (31.3) | 71 (42.8) | 21 (12.7) |
| | Overweight | 69 (79.3) | 32 (36.8) | 49 (56.3) | 48 (55.2) | 15 (17.2) |
| | Obese | 40 (95.2) | 22 (52.4) | 24 (57.1) | 34 (81.0) | 14 (33.3) |
| | **p-value** | **<0.001*** | **0.001*** | **<0.001*** | **<0.001*** | **0.012*** |
| **FBG** | Normal [a] | 150 (59.8) | 58 (23.1) | 82 (32.7) | 109 (43.4) | 39 (15.5) |
| | Pre-diabetes [b] | 36 (90.0) | 21 (52.5) | 26 (65.0) | 28 (70.0) | 7 (17.5) |
| | Diabetes [c] | 28 (93.3) | 20 (66.7) | 21 (70.0) | 22 (73.3) | 7 (23.3) |
| | **p-value** | **<0.001*** | **<0.001*** | **<0.001*** | **<0.001*** | **0.545** |
| **Blood pressure (SBP/DBP (mmHg))** | <130/85 | 90 (55.2) | 33 (20.2) | 51 (31.3) | 59 (36.2) | 22 (13.5) |
| | ≥130/85 | 124 (78.5) | 66 (41.8) | 78 (49.4) | 100 (63.3) | 31 (19.6) |
| | **p-value** | **<0.001*** | **<0.001*** | **0.001*** | **<0.001*** | **0.140** |
| **Monthly income (ranked)** | Low | 74 (60.7) | 36 (29.5) | 47 (38.5) | 53 (43.4) | 13 (10.7) |
| | Medium | 59 (70.2) | 22 (26.2) | 36 (42.9) | 39 (46.4) | 17 (20.2) |
| | High | 81 (70.4) | 41 (35.7) | 46 (40.0) | 67 (58.3) | 23 (20.0) |
| | **p-value** | **0.202** | **0.333** | **0.822** | **0.060** | **0.086** |
| Wealth index [d] | Low | 58 (54.7) | 25 (23.6) | 35 (33.0) | 45 (42.5) | 13 (12.3) |
| | Medium | 87 (77.0) | 43 (38.1) | 53 (46.9) | 66 (58.4) | 22 (19.5) |
| | High | 69 (67.6) | 31 (30.4) | 41 (40.2) | 48 (47.1) | 18 (17.6) |
| | **p-value** | **0.002*** | **0.068** | **0.112** | **0.051** | **0.333** |
| **Intensity of daily work activity** | Vigorous | 18 (56.3) | 4 (12.5) | 13 (40.6) | 11 (34.4) | 4 (12.5) |
| | Moderate | 175 (68.1) | 81 (31.5) | 102 (39.7) | 132 (51.4) | 43 (16.7) |
| | Low | 21 (65.6) | 14 (43.8) | 14 (43.8) | 16 (50.0) | 6 (18.8) |
| | **p-value** | **0.239** | **0.010*** | **0.676** | **0.281** | **0.742** |
| **Sitting time per day (ranked)** | Low | 59 (60.8) | 25 (25.8) | 35 (36.1) | 48 (49.5) | 11 (11.3) |
| | Medium | 93 (70.5) | 38 (28.8) | 51 (38.6) | 67 (50.8) | 24 (18.2) |
| | High | 62 (67.4) | 36 (39.1) | 43 (46.7) | 44 (47.8) | 18 (19.6) |
| | **p-value** | **0.370** | **0.111** | **0.293** | **0.911** | **0.250** |
| **Walking time per week in minutes** | <150 | 95 (72.5) | 47 (35.9) | 48 (36.6) | 74 (56.5) | 27 (20.6) |
| | ≥150 | 119 (62.6) | 52 (27.4) | 81 (42.6) | 85 (44.7) | 26 (13.7) |

(*Continued*)

**Table 3.** (Continued)

| Variables | Categories | Dyslipidemia (n (%)) | Elevated TC (≥200 mg/dL) (n (%)) | Elevated TG (≥150 mg/dL) (n (%)) | High LDL-C (≥130 mg/dL) (n (%)) | Low HDL-C (M/W<40/50 mg/dL) (n (%)) |
|---|---|---|---|---|---|---|
| | p-value | 0.056 | 0.105 | 0.287 | 0.038* | 0.100 |

**Note:** *significant difference,

[a] FBG: <100.0 mg/dL,

[b] FBG: 100.0–125.9 mg/dL,

[c] FBG: ≥126.0 mg/dL,

[d] ranked using Principal Component Analysis.

**Abbreviations**: DBP: Diastolic Blood Pressure, FBG: Fasting Blood Glucose, HDL-C: High-Density Lipoprotein Cholesterol, LDL-C: Low-Density Lipoprotein Cholesterol, M: Men, SBP: Systolic Blood Pressure, WC: Waist Circumference, W: Women.

change in intensity of work. The present finding is consistent with the results reported in Palestine (66.4%) [18] and South Africa (67.3%) [19]. However, the prevalence of dyslipidemia in this study is higher than the previous findings reported in Eastern Ethiopia (34.8%) [11], Africa

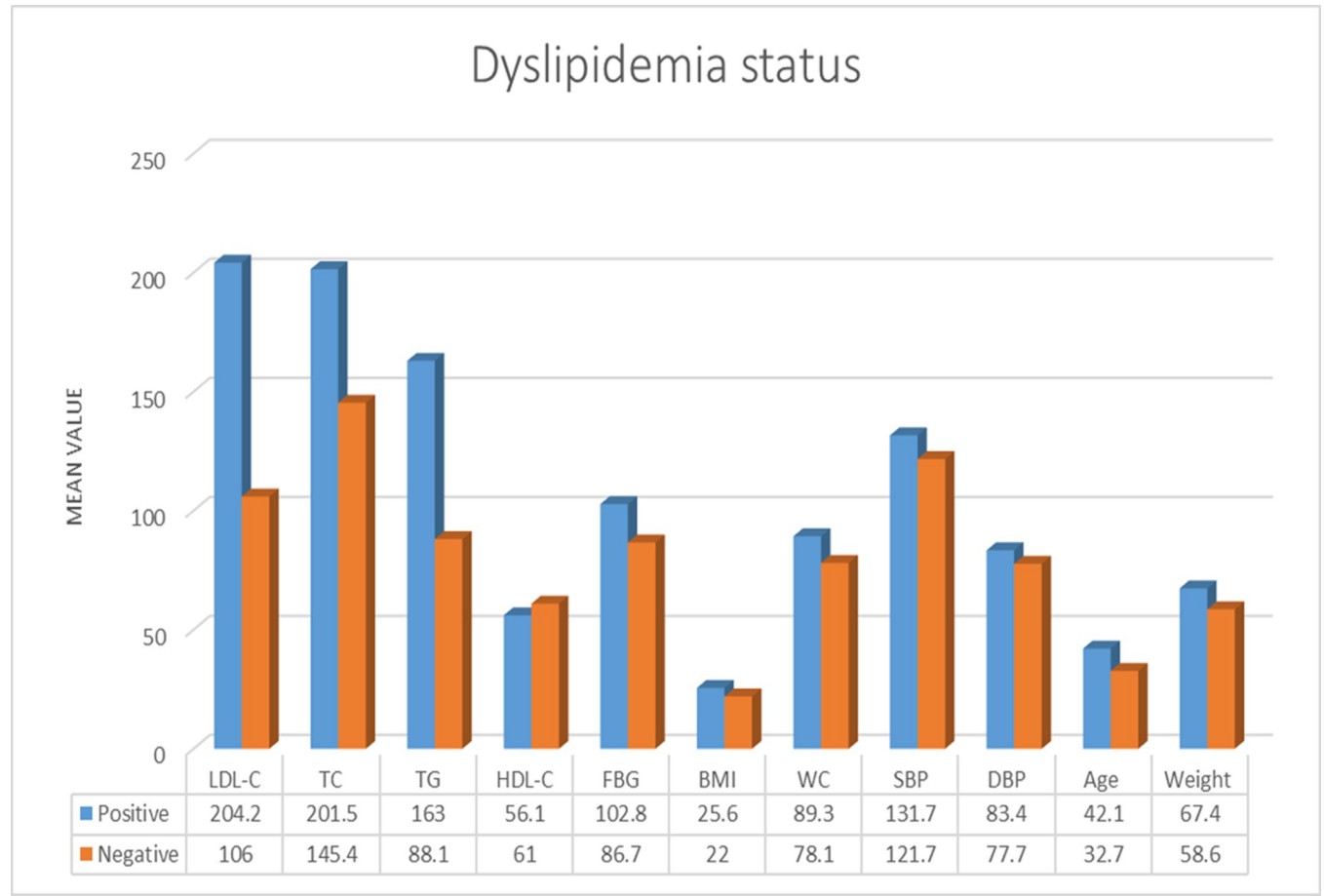

**Fig 1. Difference in the mean value of different CVD risk factors with dyslipidemia status of the participants (n = 321).** LDL-C, TC, TG, HDL-C, and FBG in mg/dl; BMI in kg/m$^2$; WC in cm; SBP and DBP in mm Hg; Age in year; Weight in kg. **Abbreviations**: BMI: Body Mass Index, DBP: Diastolic Blood Pressure, FBG: Fasting Blood Glucose, HDL-C: High-Density Lipoprotein Cholesterol, LDL-C: Low-Density Lipoprotein Cholesterol, SBP: Systolic Blood Pressure, TG: Triglyceride, TC: Total Cholesterol, WC: Waist Circumference.

**Table 4. Co-occurrence of the four lipid abnormalities stratified by gender (n = 321).**

| Lipid abnormalities | Men | Women | Total |
|---|---|---|---|
| | n (%) | n (%) | n (%) |
| **Negative** | 46 (31.7) | 61 (34.7) | 107 (33.3) |
| **TC+TG+LDL-C** | 26 (17.9) | 20 (11.4) | 46 (14.3) |
| **TC+LDL-C** | 9 (6.2) | 26 (14.8) | 35 (10.9) |
| **LDL-C** | 9 (6.2) | 18 (10.2) | 27 (8.4) |
| **TG+LDL-C** | 16 (11.0) | 8 (4.5) | 24 (7.5) |
| **TG** | 18 (12.4) | 5 (2.8) | 23 (7.2) |
| **TG+HDL-C** | 6 (4.1) | 7 (4.0) | 13 (4.1) |
| **HDL-C** | 4 (2.8) | 7 (4.0) | 11 (3.4) |
| **LDL+HDL-C** | 2 (1.4) | 8 (4.5) | 10 (3.1) |
| **TC+TG+LDL-C+HDL-C** | 1 (0.7) | 8 (4.5) | 9 (2.8) |
| **TG+LDL-C+HDL-C** | 2 (1.4) | 5 (2.8) | 7 (2.2) |
| **TC+TG** | 5 (3.4) | 0 (0.0) | 5 (1.6) |
| **TC+TG+HDL-C** | 0 (0.0) | 2 (1.1) | 2 (0.6) |
| **TC** | 1 (0.7) | 0 (0.0) | 1 (0.3) |
| **TC+LDL-C+HDL-C** | 0 (0.0) | 1 (0.6) | 1 (0.3) |
| **TC+HDL-C** | 0 (0.0) | 0 (0.0) | 0 (0.0) |

**Abbreviations**: LDL-C: Low-Density Lipoprotein Cholesterol, TC: Total Cholesterol, TG: Triglyceride, HDL-C: High-Density Lipoprotein Cholesterol.

(25.5%) [9], China (32.2%) [20], Iran (30.0%) [21], India (50.7%) [22], and Uganda (63.3%) [23]. Contrary to this, the prevalence is lower than previous studies reported in Lithuania (89.7%) [24], South Africa (85.0%) [25], India (78.4%) [26] and Poland (77.2%) [27]. This difference might be due to variation in the cutoffs, stage of urbanization in the various study settings, study period, socioeconomic status, and lifestyles of the study subjects.

High LDL-C was the most prevalent (49.5%) component of dyslipidemia followed by elevated triglyceride (40.2%), which is consistent with the previous findings reported in India [28] and China [29]. This phenomenon may reflect the growing high intake of simple carbohydrates and high saturated fat diets parallel to rapid urbanization. The prevalence of high LDL-C (49.5%) in this study is higher than the previous finding reported in Ethiopia (14.1%) [30]. It is almost similar to study findings reported in India (47.8%) [26] and Iran (50.0%) [31]. But lower than the findings reported in Thailand (56.5%) [32], Uganda (60.9%) [33], Ghana (61.0%) [34], Senegal (66.3%) [35], and Jordan (75.9%) [36]. These differences might be attributed to the variations in the cutoffs, level of urbanization, study settings, lifestyle, and socioeconomic status.

The prevalence of elevated triglyceride (40.2%) in this study is higher than the previous findings reported in Senegal (7.1%) [37], Nigeria (9.9%) [38], Ethiopia (21.0%) [30], and Malawi (28.7%) [39]. However, it is consistent with the study findings reported in Venezuela (39.7%) [40], Jordan (41.9%) [36], and Uganda (42.1%) [33]. But lower than the findings documented in Thailand (49.9%) [32], India (56.1%) [28], South Africa (59.3%) [19], and Brazil (65.3%) [41].

The prevalence of elevated total cholesterol (30.8%) in this study is almost similar to the study reported in Iran (29.6%) [21]. However, it is lower than the previous study reported in Ethiopia (33.7%) [11]. On the other hand, the prevalence of total cholesterol in this study is higher than the study findings reported in different African countries [9, 30, 33–35, 37]. The prevalence of low HDL-C (16.5%) in the present study is almost similar to the previous studies done in different African countries including Malawi (15.9%) [39], Ghana (17.0%) [34], and

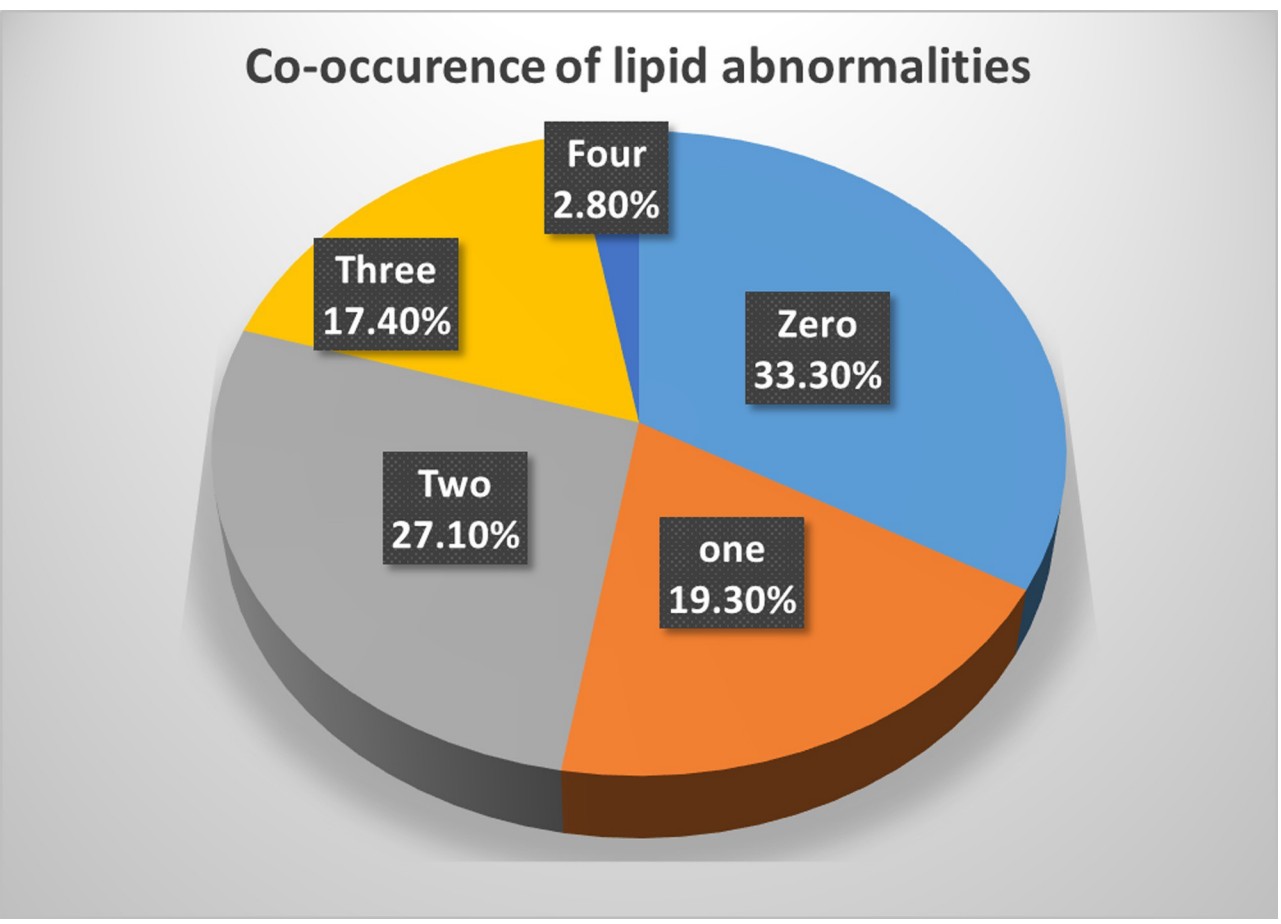

**Fig 2. The co-occurrence of lipid abnormalities and their respective proportions (n = 321).**

Africa (18.5%) [9]. Unlike many previous studies [22, 30, 38, 42–45], low HDL-C is the least prevalent component of dyslipidemia in this study.

Advanced age, higher BMI, waking less than 150 minutes per week, raised FBG, and medium socio-economic status were significantly associated with a higher risk of dyslipidemia. The prevalence of dyslipidemia markedly increased with age, peaking at the peak age range (≥65 years). The prevalence of dyslipidemia was 56.3%, 80.7%, and 86.4% among <40 years, 40–64 years, and 65 years and above, respectively. This is consistent with the findings documented elsewhere [11, 21, 46–52]. However, studies conducted in China [53] and Thailand [33] reported contradictory findings. The possible explanation for this result might be, as age increases the level of activity and intensity of work decreases, which leads to excessive fat accumulation. Besides, the socio-economic status might be improved with age, which may lead to a dietary shift.

The prevalence of dyslipidemia was also significantly increased with BMI. Around 54.7% of underweight and normal adults were dyslipidemia positive. Whereas, the prevalence was 79.3% and 95.2% among overweight and obese subjects, respectively. Many previous studies documented consistent findings [19, 23, 33, 47, 48, 51, 52, 54, 55]. This might be due to the high tendency of increasing the concentration of different lipid components as increased BMI.

An inverse relationship was observed between weekly walking time and dyslipidemia, which is similar to many previous study findings [24, 37, 46, 47, 49, 52–54]. The prevalence of

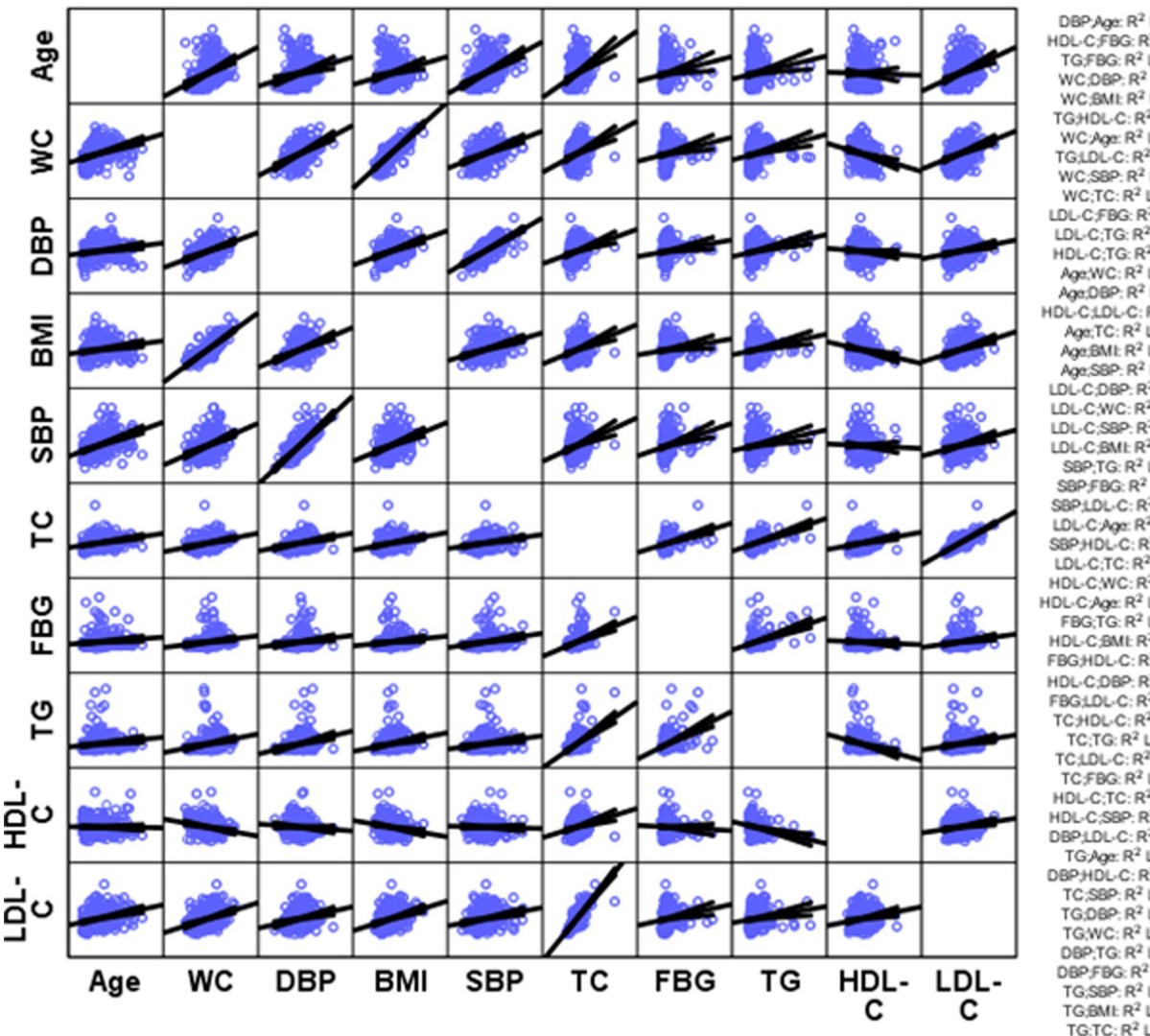

**Fig 3. Correlation of different cardiovascular disease risk factors among the participants (n = 321). Abbreviations:** BMI: Body Mass Index, DBP: Diastolic Blood Pressure, FBG: Fasting Blood Glucose, HDL-C: High-Density Lipoprotein Cholesterol, LDL-C: Low-Density Lipoprotein Cholesterol, SBP: Systolic Blood Pressure, TG: Triglyceride, TC: Total Cholesterol, WC: Waist Circumference.

dyslipidemia among subjects with a mean weekly walking time of ≥150 minutes and <150 minutes was 62.6% and 72.5%, respectively. The possible explanation for this finding might be, since the energy share of activity is increased with increasing walking time, consumed energy may not be stored in the form of lipids. Besides, stored lipids might be burned for energy to fill the energy deficit during walking, which leads to a decreased ratio of fat mass to fat-free mass.

Raised FBG was positively associated with dyslipidemia in this study. The prevalence of dyslipidemia was 59.8% and 91.4% among normal and hyperglycemic subjects, respectively. This is in line with many previous studies [11, 20, 24, 26, 32, 34, 47, 51–56]. This might be due to a close relationship between blood glucose and lipid metabolism. Because both increase with increasing body weight [57].

Medium socioeconomic status was significantly associated with a higher risk of dyslipidemia compared to low socioeconomic status. Contrary to this, subjects in the high socioeconomic status were not significantly different from subjects in the low socioeconomic status.

**Table 5. Bivariate and multivariable logistic regression analysis result (n = 321).**

| Variables | Categories | Dyslipidemia | | cOR (95%CI) | aOR (95%CI) | p-value |
|---|---|---|---|---|---|---|
| | | Yes (n (%)) | No (n (%)) | | | |
| **Sex** | Men | 99 (68.3) | 46 (31.7) | 1.142 (0.715,1.822) | | |
| | Women | 115 (65.3) | 61 (34.7) | 1 | | |
| **Age** | <40years | 107 (56.3) | 83 (43.7) | 1 | 1 | |
| | 40-64years | 88 (80.7) | 21 (19.3) | 3.251 (1.865,5.666)* | 2.196 (1.183,4.078) | 0.013* |
| | ≥65years | 19 (86.4) | 3 (13.6) | 4.913 (1.406,17.163)* | 4.334 (1.183,15.877) | 0.027* |
| **WC in cm (M/W)** | <94/80 | 86 (53.4) | 75 (46.6) | 1 | 1 | |
| | ≥94/80) | 128 (80.0) | 32 (20.0) | 3.488 (2.124,5.728)* | 1.040 (0.498,2.176) | 0.916 |
| **BMI** | Underweight & normal | 105 (54.7) | 8 (45.3) | 1 | 1 | |
| | Overweight | 69 (79.3) | 18 (20.7) | 3.176 (1.758,5.738)* | 2.500 (1.314,4.756) | 0.005* |
| | Obese | 40 (95.2) | 2 (4.8) | 16.571 (3.894,70.524)* | 15.489 (3.525,68.070) | <0.001* |
| **Educational status** | Can't read and write | 30 (85.7) | 5 (14.3) | 3.182 (1.134,8.927)* | 1.853 (0.521,6.593) | 0.341 |
| | Elementary | 36 (70.6) | 15 (29.4) | 1.273 (0.614,2.637) | 1.078 (0.455,2.554) | 0.865 |
| | Secondary | 82 (61.2) | 52 (38.8) | 0.836 (0.489,1.431) | 0.927 (0.488,1.761) | 0.816 |
| | Tertiary | 66 (65.3) | 35 (34.7) | 1 | 1 | |
| **Intensity of activity in daily work** | Vigorous | 18 (56.3) | 14 (43.7) | 1 | | |
| | Moderate | 175 (68.1) | 82 (31.9) | 1.660 (0.787,3.500) | | |
| | Low | 21 (65.6) | 11 (34.4) | 1.485 (0.541,4.077) | | |
| **Walking time per week** | <150 minutes | 95 (72.5) | 36 (27.5) | 1.574 (0.971,2.553)* | 1.722 (1.004,2.953) | 0.048* |
| | ≥150 minutes | 119 (62.6) | 71 (37.4) | 1 | 1 | |
| **Sitting time (ranked)** | Lowest | 59 (60.8) | 38 (39.2) | 1 | | |
| | Medium | 93 (70.5) | 39 (29.5) | 1.536 (0.883,2.6700) | | |
| | Highest | 62 (67.4) | 30 (32.6) | 1.331 (0.733,2.418) | | |
| **FBG** | <100.00 mg/dl | 150 (59.8) | 101 (40.2) | 1 | 1 | |
| | ≥100.00 mg/dl | 64 (91.4) | 6 (8.6) | 7.182 (2.997,17.212)* | 4.804 (1.925,11.988) | 0.001* |
| **Waist to Hip ratio (M/W)** | <0.9/0.8 | 40 (46.0) | 47 (54.0) | 1 | 1 | |
| | ≥0.9/0.8) | 174 (74.4) | 60 (25.6) | 3.408 (2.039,5.695)* | 1.423 (0.755,2.684) | 0.275 |
| **Blood pressure** | Normotensive | 90 (55.2) | 73 (44.8) | 1 | 1 | |
| | Raised blood pressure | 124 (78.5) | 34 (21.5) | 2.958 (1.814,4.825)* | 1.599 (0.887,2.881) | 0.118 |
| **Heart rate (ranked)** | Low | 70 (63.1) | 41 (36.9) | 1 | | |
| | Medium | 67 (65.0) | 36 (35.0) | 1.090 (0.623,1.907) | | |
| | High | 77 (72.0) | 30 (28.0) | 1.503 (0.849,2.662) | | |
| **Fruit intake (at least per week)** | Yes | 129 (65.5) | 68 (34.5) | 1 | | |
| | No | 85 (68.5) | 39 (31.5) | 1.149 (0.711,1.856) | | |
| **Vegetable intake (at least per week)** | Yes | 156 (65.5) | 82 (24.5) | 1 | | |
| | No | 58 (69.9) | 25 (30.1) | 1.219 (0.711,2.092) | | |
| **Formal exercise** | Yes | 26 (61.9) | 16 (38.1) | 1 | | |
| | No | 188 (67.4) | 91 (32.6) | 1.271 (0.650,2.487) | | |
| **Wealth index** | Low | 58 (54.7) | 48 (45.3) | 1 | 1 | |
| | Medium | 87 (77.0) | 26 (23.0) | 2.769 (1.548,4.953)* | 2.017 (1.044,3.899) | 0.037* |
| | High | 69 (67.6) | 33 (32.4) | 1.730 (0.984,3.042) | 1.350 (0.716,2.545) | 0.353 |
| **House servant** | Yes | 60 (77.9) | 17 (22.1) | 2.063 (1.134,3.751)* | 1.088 (0.461,2.568) | 0.847 |
| | No | 154 (63.1) | 90 (36.9) | 1 | 1 | |
| **Laundry machine** | Yes | 74 (77.1) | 22 (22.9) | 2.018 (1.167,3.489)* | 1.055 (0.517,2.154) | 0.883 |
| | No | 140 (62.5) | 84 (37.5) | 1 | 1 | |
| **House ownership** | Yes | 116 (72.0) | 45 (28.0) | 1.631 (1.021,2.606)* | 0.704 (0.377,1.314) | 0.270 |
| | No | 98 (61.3) | 62 (38.7) | 1 | 1 | |

*(Continued)*

**Table 5.** (Continued)

| Variables | Categories | Dyslipidemia | | cOR (95%CI) | aOR (95%CI) | p-value |
|---|---|---|---|---|---|---|
| | | Yes (n (%)) | No (n (%)) | | | |
| **Type of oil** | Liquid | 109 (68.1) | 51 (31.9) | 1.140 (0.716,1.814) | | |
| | Solid | 105 (65.2) | 56 (34.8) | 1 | | |
| **Alcohol consumption** | Yes | 169 (68.1) | 79 (31.9) | 1.315 (0.743,2.325) | | |
| | No | 45 (61.6) | 28 (38.4) | 1 | | |

**Note:** Maximum SE: 0.671; Hosmer-Lemeshow: 0.572,

*significant association.

**Abbreviations:** cOR: crude Odds Ratio; aOR: adjusted Odds Ratio, FBG: Fasting Blood Glucose, BMI: Body Mass Index, WC: Waist Circumference, M: Men, W: Women.

The prevalence of dyslipidemia among subjects with low, medium, and high socioeconomic status was 54.7%, 77.0%, and 67.6%, respectively, which is in line with the previous study reported in China [51]. This might be related to better economic access to alcoholic drinks, energy-dense foods, refined carbohydrates, and physical inactivity. The poor cannot afford energy-dense foods and are engaged in energy-demanding daily work, and the rich can afford healthy foods.

The prevalence of dyslipidemia among individuals who had house servant (77.9%), laundry machine (77.1%), and private house (72.0%) was higher than in individuals who had no house servant (63.1%), laundry machine (62.5%), and private house (61.3%). If an individual has a house servant or laundry machine, s/he may stop household chores. This may lead to physical inactivity, less energy expenditure, and more weight gain. Similarly, house ownership may be associated with better economic access to energy-dense foods, physical inactivity, engagement in low-intensity work, and low energy expenditure. This may cause weight gain and accumulation of excess fat, which leads to dyslipidemia. However, the effect of having a house servant, laundry machine, and house ownership on dyslipidemia was not statistically significant in this study.

As a limitation, the prevalence of dyslipidemia was based on a single laboratory test, which may lead to minor inaccuracies. As all cross-sectional study designs, limits the ability to address causal relationships between dyslipidemia and its identified associated risk factors. Since the data were collected through a questionnaire, this may lead to a recall bias.

## Conclusion

In this study, the prevalence of dyslipidemia and its lipid components particularly high LDL-C, elevated triglyceride, and elevated total cholesterol were unacceptably high. Advanced age, increased BMI, walking less than 150 minutes per week, hyperglycemia, and medium socioeconomic status were significantly associated with increased risk of dyslipidemia. All are modifiable risk factors except age. This result highlights an urgent need to develop and implement appropriate intervention programs aimed at controlling the risk factors and introducing routine screening programs in the urban areas of Ethiopia. Besides, it is necessary to improve the awareness of individuals on the risk factors, and the use of proper therapeutics like nutritional, exercise, and behavioral interventions.

## Supporting information

**S1 File. Used dataset for dyslipidemia 2020.**
(SAV)

## Acknowledgments

We would like to express our heartfelt gratitude to all study participants, data collectors, Tigray Health Research Institute, Jimma University, and Adigrat University for their support.

## Author Contributions

**Conceptualization:** Gebremedhin Gebreegziabiher.

**Data curation:** Gebremedhin Gebreegziabiher, Kibrti Mehari.

**Formal analysis:** Gebremedhin Gebreegziabiher, Tefera Belachew, Kibrti Mehari, Dessalegn Tamiru.

**Funding acquisition:** Gebremedhin Gebreegziabiher.

**Investigation:** Gebremedhin Gebreegziabiher, Tefera Belachew, Kibrti Mehari, Dessalegn Tamiru.

**Methodology:** Gebremedhin Gebreegziabiher, Tefera Belachew, Kibrti Mehari, Dessalegn Tamiru.

**Project administration:** Gebremedhin Gebreegziabiher.

**Resources:** Gebremedhin Gebreegziabiher.

**Software:** Gebremedhin Gebreegziabiher, Dessalegn Tamiru.

**Supervision:** Gebremedhin Gebreegziabiher, Kibrti Mehari.

**Validation:** Gebremedhin Gebreegziabiher, Tefera Belachew, Dessalegn Tamiru.

**Visualization:** Gebremedhin Gebreegziabiher.

**Writing – original draft:** Gebremedhin Gebreegziabiher, Tefera Belachew, Kibrti Mehari, Dessalegn Tamiru.

**Writing – review & editing:** Gebremedhin Gebreegziabiher, Tefera Belachew, Kibrti Mehari, Dessalegn Tamiru.

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
