## [Decision Letter · Decision Letter 0]

25 Aug 2020

PONE-D-20-23153

Prevalence of dyslipidemia and associated risk factors among adults living in Mekelle city, Northern Ethiopia, a community-based cross-sectional study.

PLOS ONE

Dear Dr. Gebrehiwot,

Thank you for submitting your manuscript to PLOS ONE. After careful consideration, we feel that it has merit but does not fully meet PLOS ONE’s publication criteria as it currently stands. Therefore, we invite you to submit a revised version of the manuscript that addresses the points raised during the review process.

Two experts in the field handled your manuscript, and we are very thankful for their time and efforts. Although some interest was found in your study, there were major concerns that arose during review that overshadowed this enthusiasm. Notably, there were several comments about the experimental design, including the need for describing the inclusion criteria; the data presentation needs work; and the text needs to be rewritten to simplify the results and explanations. Someone should be hired to proof the manuscript for spelling and grammar. Please address ALL of the reviewers' comments in your revised manuscript.

We look forward to receiving your revised manuscript.

Kind regards,

Frank T. Spradley

Academic Editor

PLOS ONE

2. Your ethics statement must appear in the Methods section of your manuscript. If your ethics statement is written in any section besides the Methods, please move it to the Methods section and delete it from any other section. Please also ensure that your ethics statement is included in your manuscript, as the ethics section of your online submission will not be published alongside your manuscript.

Reviewers' comments:

Reviewer's Responses to Questions

**Comments to the Author**

1. Is the manuscript technically sound, and do the data support the conclusions?

Reviewer #1: No

Reviewer #2: Yes

2. Has the statistical analysis been performed appropriately and rigorously? 

Reviewer #1: No

Reviewer #2: Yes

3. Have the authors made all data underlying the findings in their manuscript fully available?

Reviewer #1: Yes

Reviewer #2: Yes

4. Is the manuscript presented in an intelligible fashion and written in standard English?

Reviewer #1: No

Reviewer #2: Yes

5. Review Comments to the Author

Reviewer #1: 1- Given the definition of the prevalence of the disease in a community, it cannot be said that this study examined the prevalence of dyslipidemia in Ethiopia.

2- What was the inclusion criteria?

3- From line 219 to line 241 the text is added because most of the explanations are in Table 1 and there is no need for explanation in the text.

More appropriate and concise sentences can be used instead. For example, most people were married and the marital status of men and women was significantly different (P = 0.001), women consumed significantly more vegetables during the week than men (P = 0.007).

4- In line 204, the name of the test is Hosmer, not Hesmer.

5- Many parts of an article have a very long text.

Reviewer #2: This manuscript provides critical information on dyslipidemic rates in the Northern Ethiopian population. Overall, the study was well designed and written, but a few points should be addressed.

1). Carefully check for spelling errors and proper grammatical usage throughout the manuscript and correct. Lines 124-128 read somewhat disjointed and are confusing to the reader (possible footnotes for further description?). Please clarify this information.

2). Under Methods: Move ' Ethical Issues' subsection information into the 'Study Design and Participants' subsection at the beginning of the Methods section.

3) Considering that roughly ~10% of participants could not read/write, how was consent given from these participants? Was proxy or next-of-kin consent given for these participants? Were they read the consent form specifically? Mention how the study gained consent in this subpopulation and list this important detail in the Methods.

4). While table 3 does a great job addressing the covariable relationship between several characteristics of the participants and dyslipidemia, an interesting comparison should be made to include "Alcohol use" which was significantly greater in the cohort. Because alcohol can dramatically alter lipid metabolism, this comparison should be made.

5). TABLE 1: The far left column is confusing to read. Recommend a headline for the far left column as "Characteristics" and show each characteristic in the subcolumn either outlined with separate borders or place break lines between characteristics. As written, the wording seems to run together and is difficult to distinguish transitions to the next characteristic listed.

6) TABLE 3: Add borders around the p-value row. As mentioned above, it is difficult to identify and read the table as presented. Consider alternative ways to organize the data, or at the very least, add more break lines and subcolumns to organize the presentation in a clearer manner.

7). Discussion: Line 453: Discuss how characteristics related to household income (e.g. House ownership, wealth, house servant, laundry machine) are related to the prevalence of dyslipidemia. Also, significant alcohol use in the cohort is a limitation that should be addressed.

6. PLOS authors have the option to publish the peer review history of their article (what does this mean?). If published, this will include your full peer review and any attached files.

Reviewer #1: No

Reviewer #2: No

---

## [Author Response · Author response to Decision Letter 0]

22 Sep 2020

Editor's comments

Please ensure that your manuscript meets PLOS ONE's style requirements:- Manuscript prepared according to PLOS ONE's style.

Your ethics statement must appear in the Methods section of your manuscript:- Ethics statement incorporated in the Methods and materials section. 

Reviewer 1

1. Given the definition of the prevalence of the disease in a community, it cannot be said that this study examined the prevalence of dyslipidemia in Ethiopia:- We admit the comment and addressed as follows. 

This study was aimed to estimate the prevalence of dyslipidemia and its associated risk factors among adults residing in Mekelle city, Northern Ethiopia.

2. What was the inclusion criteria?:- Being an adult aged 20 years and above and residents who lived at least 6 months in the city were considered as the inclusion criteria. 

3. From line 219 to line 241 the text is added because most of the explanations are in Table 1 and there is no need for explanation in the text. More appropriate and concise sentences can be used instead. For example, most people were married and the marital status of men and women was significantly different (P = 0.001), women consumed significantly more vegetables during the week than men (P = 0.007):- Addressed as recommended. We have indicated only the most important findings from table 1 in the text.

4. In line 204, the name of the test is Hosmer, not Hesmer:- Addressed as recommended. The goodness of fit of the model was checked using the Hosmer-Lemeshow test at p>0.05.

5. Many parts of an article have a very long text:- We make the article short and precise without missing the most important content.

Reviewer 2

1. Carefully check for spelling errors and proper grammatical usage throughout the manuscript and correct. Lines 124-128 read somewhat disjointed and are confusing to the reader (possible footnotes for further description?). Please clarify this information:- We made extensive revision for spelling and grammatical errors and modified accordingly. 

The information in Lines 124-128 is clarified as recommended.

2. Under Methods: Move ' Ethical Issues' subsection information into the 'Study Design and Participants' subsection at the beginning of the Methods section:- ‘Ethical issues’ moved to the recommended subsection (under study design, setting, and participants).

3. Considering that roughly ~10% of participants could not read/write, how was consent given from these participants? Was proxy or next-of-kin consent given for these participants? Were they read the consent form specifically? Mention how the study gained consent in this subpopulation and list this important detail in the Methods:- Data collectors read the information to each illiterate participant and asked him or her for voluntary participation. If voluntary, they put their fingerprint as a signature in the written consent form. This is the usual practice in developing countries, where many people are illiterate. This information is indicated in the Ethical issues subsection under materials and methods.

4. While table 3 does a great job addressing the covariable relationship between several characteristics of the participants and dyslipidemia, an interesting comparison should be made to include "Alcohol use" which was significantly greater in the cohort. Because alcohol can dramatically alter lipid metabolism, this comparison should be made:- We have included alcohol use in Table 3 and Table 5 according to reviewer’s recommendation. However, the effect is not statistically significant as expected. The possible reason might be the frequency, type, and amount of alcohol consumed matters. Since alcohol is expensive, most people in the study area consumed few numbers of bottle of beer per week, which may not have significant effect on the lipid profile of individuals. 

5. TABLE 1: The far left column is confusing to read. Recommend a headline for the far left column as "Characteristics" and show each characteristic in the subcolumn either outlined with separate borders or place break lines between characteristics. As written, the wording seems to run together and is difficult to distinguish transitions to the next characteristic listed:- Addressed as recommended. Two separate columns have been created (named as ‘variables’ and ‘categories’).

6. TABLE 3: Add borders around the p-value row. As mentioned above, it is difficult to identify and read the table as presented. Consider alternative ways to organize the data, or at the very least, add more break lines and subcolumns to organize the presentation in a clearer manner:- Table 3 has been re-organized as recommended. 

7. Discussion: Line 453: Discuss how characteristics related to household income (e.g. House ownership, wealth, house servant, laundry machine) are related to the prevalence of dyslipidemia. Also, significant alcohol use in the cohort is a limitation that should be addressed:- Having better household assets is related with better access to energy dense foods, physical inactivity, engagement in low intensity work, low energy expenditure all of which may lead to weight gain and accumulation of excessive fat, which in turn may lead to dyslipidemia. 

More than one-quarters of participants in the present study were alcohol consumers and we tried to assess the association with dyslipidemia according to your recommendation. However, the effect is not significant as expected. The possible reason might be the frequency, type, and amount of alcohol consumed matters. Since alcohol is expensive, most people consume few numbers of bottle of beer per week, which may not have significant effect on the lipid profile of individuals.

NB A rebuttal letter that responds to each point raised by the academic editor and reviewers is uploaded together with other necessary documents.

---

## [Decision Letter · Decision Letter 1]

14 Oct 2020

PONE-D-20-23153R1

Prevalence of dyslipidemia and associated risk factors among adults living in Mekelle city, Northern Ethiopia.

PLOS ONE

Dear Dr. Gebrehiwot,

Thank you for submitting your manuscript to PLOS ONE. After careful consideration, we feel that it has merit but does not fully meet PLOS ONE’s publication criteria as it currently stands. Therefore, we invite you to submit a revised version of the manuscript that addresses the points raised during the review process.

There are still major revisions that require the attention of the authors. Notably, these revisions relate to English grammar and syntax. The reviewer has kindly offered suggestions in this regard, but the manuscript must be critically reviewed by the authors for readability. Moreover, and importantly, there are concerns about discrepancies in the findings. Please address ALL of the reviewer's comments in your revised manuscript.

We look forward to receiving your revised manuscript.

Kind regards,

Frank T. Spradley

Academic Editor

PLOS ONE

Reviewers' comments:

Reviewer's Responses to Questions

**Comments to the Author**

1. If the authors have adequately addressed your comments raised in a previous round of review and you feel that this manuscript is now acceptable for publication, you may indicate that here to bypass the “Comments to the Author” section, enter your conflict of interest statement in the “Confidential to Editor” section, and submit your "Accept" recommendation.

Reviewer #1: (No Response)

2. Is the manuscript technically sound, and do the data support the conclusions?

Reviewer #1: Yes

3. Has the statistical analysis been performed appropriately and rigorously? 

Reviewer #1: Yes

4. Have the authors made all data underlying the findings in their manuscript fully available?

Reviewer #1: Yes

5. Is the manuscript presented in an intelligible fashion and written in standard English?

Reviewer #1: (No Response)

6. Review Comments to the Author

Reviewer #1: Generally

Although the statistical methods used in this article are good and the analysis is well done, but is poorly written in terms of the principles of article writing.

I have mentioned many corrections in the findings section and other parts of the findings also need corrections similar to the ones mentioned, and this type of bug has been repeated throughout the findings section. The authors need to modify and resubmit the entire section of the findings as noted

......................................................................

It is best to add a reference to line 102, To determine what is the formula for calculating the sample size.

The “Materials and methods” section is very long and should be summarized. For example, if possible,” Study design and setting” And “Study population”, it is better to combine them and write more briefly.

Also, if possible,” Selection and training sections of data collectors” And “Data collection and quality control”, it is better to combine them and write more briefly.

Also, if possible,” Blood sample collection and laboratory analysis” And “Definition of terms”, it is better to combine them and write more briefly.

Please shorten the text wherever possible in all parts of the manuscript. For example in line 172, the phrase “Sample characteristics were expressed as mean ± standard deviation (SD) for continuous variables, and frequencies and percentages for categorical variables” can be omitted because this is specified in the results section.

In “Result” section, As mentioned in the previous review, items that are clear from the tables should not be repeated in the text. Please provide brief explanations in the text only in cases where the difference is significant.

It is better to remove the phrase “the mean + SD age of the participants was 38.99 + 14.22 years” in line 197.

It is better to remove the phrase “More than half (54.8%) of participants were women.” in line 197.

In line 199 to 203, additional text has been written about marital status, occupation, smoking and alcohol, and a brief explanation can be written instead. For example, please briefly explain that men and women were significantly different in terms of marriage, occupation, smoking and alcohol (P<0.005). And only give a brief explanation if you see an interesting result for each of them.

Please correct lines 203 to 210 as described above.

In line 207, in the table of this frequency, 13.1 is written, but in the text, 13.4.

In line 221, the phrase "slightly higher among men (68.3%) compared to women (65.3%)" It is better to remove it because it is not more common in men.

Please delete these phrases because they are clear from the table:

Line 222 “Half (49.5%) of the study participants had elevated LDL-C while 30.8% had an elevated total cholesterol level”

Line 226 “More than one fourth (27.1%) of the participants were overweight.”

Line 230 “Almost one-thirds (30.5%) and 18.7% of the participants were hypertensive and pre-hypertensive respectively”

Line 233 “The mean + SD BMI and WC were 24.4+4.9 kg/m2 and 85.6+13.6 cm respectively, with no significant difference across gender”

Line 259 “The prevalence of dyslipidemia was 57.8%, 79.3%, and 95.2% among normal, overweight, and obese participants respectively”

Also is in lines 260 to 264.

In line 229, it is better to write that the difference between men and women was significant.

In line 257 It cannot be said that the behavior of TG and HDL with the age variable, is as mentioned.

In line 264, this phrase is the type of phrase at the beginning of the paragraph. It is better to combine them and bring them in one sentence: "The risk of dyslipidemia and its component lipid abnormalities were also consistently increased with FBG and blood pressure."

In Table 2, there is a discrepancy in the findings! For the “Waist to hip ratio” in qualitative state, it is observed that a higher percentage of women have high values of Waist to hip ratio. But in quantitative state, we see that the average of this variable is higher in men. It’s not logical.

7. PLOS authors have the option to publish the peer review history of their article (what does this mean?). If published, this will include your full peer review and any attached files.

Reviewer #1: No

---

## [Author Response · Author response to Decision Letter 1]

3 Nov 2020

Response letter

Dear Editor:

I have sent the revised manuscript entitled “Prevalence of dyslipidemia and associated risk factors among adult residents of Mekelle City, Northern Ethiopia.” I really appreciate for your critical review process. All the raised comments are very helpful in improving the manuscript. We have addressed all the comments raised by an editor and the reviewer in the manuscript and we have summarized in the following table. Besides, we tried to shorten the whole manuscript and major revision has been made on grammar and syntax.

1. It is best to add a reference to line 102, To determine what is the formula for calculating the sample size:- 

Lwanga S, Lemeshow S. Sample size determination in health studies: a practical manual. England: World Health Organization; 1991. (Page 5, Line 102)

2. The “Materials and methods” section is very long and should be summarized. For example, 

if possible,” Study design and setting” And “Study population”, it is better to combine them and write more briefly:- 

We have revised the Materials methods section. We have combined and briefly wrote the sub-sections based on your recommendation. As much as possible we make it short and precise. (Page 4-5, Line 84-90)

if possible,” Selection and training sections of data collectors” And “Data collection and quality control”, it is better to combine them and write more briefly:- 

We have combined and briefly wrote the sub-sections based on your recommendation. (Page 6-7, Line 116-137)

if possible,” Blood sample collection and laboratory analysis” And “Definition of terms”, it is better to combine them and write more briefly:_

We have combined and briefly wrote the sub-sections based on your recommendation. (Page 7-8, Line 140-158)

3. Please shorten the text wherever possible in all parts of the manuscript. For example in line 172, the phrase “Sample characteristics were expressed as mean ± standard deviation (SD) for continuous variables, and frequencies and percentages for categorical variables” can be omitted because this is specified in the results section:- 

The indicated phrase is deleted from the data analysis part. Besides, the whole manuscript is revised to minimize the content. Unnecessary repetitions have been deleted. (Page 8, Line 161-171)

4. In “Result” section, As mentioned in the previous review, items that are clear from the tables should not be repeated in the text. Please provide brief explanations in the text only in cases where the difference is significant. It is better to remove the phrase “the mean + SD age of the participants was 38.99 + 14.22 years” in line 197. It is better to remove the phrase “More than half (54.8%) of participants were women.” in line 197.

Unnecessary phrases are deleted from all table descriptions based on your recommendation. Explanation is provided only for variables with significant difference throughout the manuscript. (Page 9, Line 179-185)

5. In line 199 to 203, additional text has been written about marital status, occupation, smoking and alcohol, and a brief explanation can be written instead. 

For example, please briefly explain that men and women were significantly different in terms of marriage, occupation, smoking and alcohol (P<0.005). 

We have addressed the comment as you have recommended. (page 9, Line 179-180)

And only give a brief explanation if you see an interesting result for each of them. Please correct lines 203 to 210 as described above.

We have addressed the comment as you have recommended. (Page 9, Line 179-185)

6. In line 207, in the table of this frequency, 13.1 is written, but in the text, 13.4.

Thank you for your critical observation. We have corrected it. (13.1%) (Page 9, Line 183)

7. In line 221, the phrase "slightly higher among men (68.3%) compared to women (65.3%)" It is better to remove it because it is not more common in men.

Removed as recommended. (Page 10, Line 192-197)

8. Please delete these phrases because they are clear from the table:

Line 222 “Half (49.5%) of the study participants had elevated LDL-C while 30.8% had an elevated total cholesterol level”

Line 226 “More than one fourth (27.1%) of the participants were overweight.”

Line 230 “Almost one-thirds (30.5%) and 18.7% of the participants were hypertensive and pre-hypertensive respectively”

Line 233 “The mean + SD BMI and WC were 24.4+4.9 kg/m2 and 85.6+13.6 cm respectively, with no significant difference across gender”

Line 259 “The prevalence of dyslipidemia was 57.8%, 79.3%, and 95.2% among normal, overweight, and obese participants respectively”

Also is in lines 260 to 264.

All phrases indicated in your comment have been deleted based on your recommendation. We really appreciate your critical review. (Page 10, Line 192-197)

9. In line 229, it is better to write that the difference between men and women was significant.

Addressed (Page 10, Line 193-195)

10. In line 257 It cannot be said that the behavior of TG and HDL with the age variable, is as mentioned.

Though the difference in TG was significant with age, there was inconsistency. Whereas, HDL-C was not significantly different with age. Addressed as recommended. (Page 11, Line 215-216)

11. In line 264, this phrase is the type of phrase at the beginning of the paragraph. It is better to combine them and bring them in one sentence: "The risk of dyslipidemia and its component lipid abnormalities were also consistently increased with FBG and blood pressure."

We have combined the two sentences as you have commented. (Page 11, Line 212-214)

12. In Table 2, there is a discrepancy in the findings! For the “Waist to hip ratio” in qualitative state, it is observed that a higher percentage of women have high values of Waist to hip ratio. But in quantitative state, we see that the average of this variable is higher in men. It’s not logical.

This is due to the difference in the cutoff point. Men have greater cutoff (0.9) than women (0.8). The same is true for waist circumference, with a cutoff 94 cm and 80 cm for men and women, respectively. Besides, women have naturally greater hip circumference relative to their waist circumference due to excessive fat accumulation around the hip compared to men, which makes women to have lower cutoff. Therefore, significantly higher proportion of women have raised waist to hip ratio (>0.8) and men have significantly higher mean value of waist to hip ratio. (Page 10, Line 193-195)

Sincerely yours.

Gebremedhin Gebreegziabiher (corresponding author)

Email: ghingherg@gmail.com, 

Mobile: +251914754562, 

Nov-03/2020

---

## [Editor Report · Decision Letter 2]

16 Nov 2020

Prevalence of dyslipidemia and associated risk factors among adult residents of Mekelle City, Northern Ethiopia

PONE-D-20-23153R2

Dear Dr. Gebrehiwot,

We’re pleased to inform you that your manuscript has been judged scientifically suitable for publication and will be formally accepted for publication once it meets all outstanding technical requirements.

Kind regards,

Frank T. Spradley

Academic Editor

PLOS ONE

---

## [Editor Report · Acceptance letter]

21 Dec 2020

PONE-D-20-23153R2 

Prevalence of dyslipidemia and associated risk factors among adult residents of Mekelle City, Northern Ethiopia 

Dear Dr. Gebreegziabiher:

I'm pleased to inform you that your manuscript has been deemed suitable for publication in PLOS ONE. Congratulations! Your manuscript is now with our production department. 

Kind regards, 

on behalf of

Dr. Frank T. Spradley 

Academic Editor

PLOS ONE